# 27 T ultra-high static magnetic field changes orientation and morphology of mitotic spindles in human cells

Lei Zhang[1,2†], Yubin Hou[1†], Zhiyuan Li[1], Xinmiao Ji[1], Ze Wang[1,2], Huizhen Wang[1,2], Xiaofei Tian[1,2], Fazhi Yu[2], Zhenye Yang[2], Li Pi[1,2], Timothy J Mitchison[3*], Qingyou Lu[1,2,4*], Xin Zhang[1*]

[1]High Magnetic Field Laboratory, Chinese Academy of Sciences, Hefei, China; [2]University of Science and Technology of China, Hefei, China; [3]Department of Systems Biology, Harvard Medical School, Boston, United States; [4]Collaborative Innovation Center of Advanced Microstructure, Nanjing University, Nanjing, China

**Abstract** Purified microtubules have been shown to align along the static magnetic field (SMF) in vitro because of their diamagnetic anisotropy. However, whether mitotic spindle in mammalian cells can be aligned by magnetic field has not been experimentally proved. In particular, the biological effects of SMF of above 20 T (Tesla) on mammalian cells have never been reported. Here we found that in both CNE-2Z and RPE1 human cells spindle orients in 27 T SMF. The direction of spindle alignment depended on the extent to which chromosomes were aligned to form a planar metaphase plate. Our results show that the magnetic torque acts on both microtubules and chromosomes, and the preferred direction of spindle alignment relative to the field depends more on chromosome alignment than microtubules. In addition, spindle morphology was also perturbed by 27 T SMF. This is the first reported study that investigated the mammalian cellular responses to ultra-high magnetic field of above 20 T. Our study not only found that ultra-high magnetic field can change the orientation and morphology of mitotic spindles, but also provided a tool to probe the role of spindle orientation and perturbation in developmental and cancer biology.

**\*For correspondence:**
timothy_mitchison@hms.harvard.edu (TJM); qxl@ustc.edu.cn (QL); xinzhang@hmfl.ac.cn (XZ)

[†]These authors contributed equally to this work

**Competing interests:** The authors declare that no competing interests exist.

## Introduction

The mitotic spindle is a highly dynamic microtubule assembly responsible for chromosome segregation and cleavage furrow positioning during cell division. Spindle orientation is central to cell fate determination and tissue architecture, and mounting evidences show that astral microtubules, multiple cortical proteins and their interactions with plasma membrane are all critical to achieve correct orientation (*Toyoshima et al., 2007*; *Lancaster and Knoblich, 2012*; *Lu and Johnston, 2013*; *Bergstralh and St Johnston, 2014*; *Nestor-Bergmann et al., 2014*). Tissues may have additional control mechanisms to minimize spindle orientation errors, such as apoptosis in the wing disc (*Bergstralh and St Johnston, 2014*). The connection between spindle orientation and tumorigenesis has also been an active field in recent years (*Gonzalez, 2007*; *Knoblich, 2010*; *Pease and Tirnauer, 2011*).

Most biological materials are diamagnetic, such as proteins, DNA and lipids. SMFs can align large biological objects that have diamagnetic anisotropy, such as microtubule polymers and nucleic acid chains, as well as some types of cells and organisms (*Maret and Dransfeld, 1977*; *Torbet and Ronzière, 1984*; *Higashi et al., 1993*; *Emura et al., 2001*; *Takeuchi et al., 2002*; *Teodori et al., 2006*; *Stern-Straeter et al., 2011*). The degree of alignment with the externally applied magnetic field is proportional to the product of the molecular magnetic susceptibility and the magnetic field strength.

**eLife digest** Nowadays, a number of methods can be used to 'look' inside the body to investigate potential health problems. One of these is a technique called magnetic resonance imaging (MRI) that uses magnetic fields that are several hundred times stronger than a fridge magnet (or over 10,000 times stronger than the Earth's natural magnetic field) to generate images of the inside of the body. In general, stronger magnetic fields enable higher quality images to be obtained. However, the effects of exposing the body's cells to these magnetic fields have not been fully determined.

Like most other biological materials, protein polymers called microtubules can respond to high magnetic fields – for example, by aligning with the field. Microtubules play a number of roles inside cells. This includes forming the mitotic spindle that separates copies of chromosomes – the structures in which the majority of a cell's genetic material is stored – equally between dividing cells.

The orientation of the mitotic spindle determines the direction in which a cell will divide. This direction is important for generating different types of cells and tissues. Furthermore, many cancerous cells have incorrectly oriented spindles.

Zhang, Hou et al. have now exposed cancerous and normal human cells to magnetic fields of varying strengths. The maximum magnetic field strength tested (27 Tesla – or around 10 times the highest field strengths produced by standard hospital MRI scanners) did not kill the cells after four hours of exposure, but the orientation of the spindles inside the cells did change. In addition, the 27 Tesla magnetic field caused spindles that were perpendicular to the direction of the field to widen. At an intermediate field strength (9 Tesla – a magnetic field strength that has been used in some experimental MRI scanners), the orientation of the spindle only changed after three days of continuous exposure to the magnetic field. Lower field strengths (such as those currently used in hospital MRI scanners) did not alter the orientation of the spindle even after seven days of exposure.

Zhang, Hou et al. also observed that the magnetic field acts on both the microtubules and chromosomes. However, the alignment of the chromosomes in the cell was the greatest determinant of the direction in which the spindle would align itself in response to the magnetic field.

The next step is to analyze the consequences of magnetic field-induced spindle orientation changes – can these lead to cancer or reduce cancer growth, or change how animal tissues develop? Understanding how to control the position of the spindle could also ultimately make it possible to use ultra-high magnetic fields to engineer tissues or stimulate their regeneration.

For proteins, the diamagnetic anisotropy is mainly due to the alpha helix, beta sheet and aromatic rings (*Chabre, 1978*; *Worcester, 1978*; *Pauling, 1979*). Even individual peptide bonds, which have weak diamagnetic anisotropy can contribute when linked together in a fixed and organized orientation in alpha helix or beta sheet. Most proteins have very weak diamagnetic anisotropy, but their response to a magnetic field can be amplified by ordered polymerization, as in microtubules, where the additive diamagnetic anisotropy can be significant. The DNA chain is another example of a biopolymer with relatively large diamagnetic anisotropy (*Maret et al., 1975*), mainly due to its stacked aromatic bases. Theoretical predictions suggested that mitotic chromosome arms, where DNA is highly compacted, might generate electromagnetic fields along the chromosome arm direction (*Zhao and Zhan, 2012*) and, less speculatively, that chromosomes should be fully aligned by SMFs of around 1.4 T (*Maret, 1990*).

Mitotic spindles comprise both microtubules and chromosomes, but most studies of their potential response to magnetic fields have focused on microtubules. Multiple studies have shown that purified microtubules can be aligned by moderate and high SMFs and the alignment effect increases significantly with magnetic field strength (*Vassilev et al., 1982*; *Bras et al., 1998*; *Glade and Tabony, 2005*). This is due to the diamagnetic anisotropy of tubulin and microtubules (*Bras et al., 2014*). In addition, tubulin assembly in vitro was disordered by a 10–100 nT hypogeomagnetic field (natural geomagnetic field is usually around 50000 nT/0.5 Gauss) (*Wang et al., 2008*). However, whether SMF can change the mitotic spindle orientation in a cell has never been reported. Denegre et al found that 16.7 T large gradient SMFs can affect the division orientation of Xenopus eggs

(*Denegre et al., 1998*). They proposed that SMF may affect the orientation of astral microtubules and/or spindles, which was theoretically and experimentally proved later (*Valles, 2002*; *Valles et al., 2002*). Although the importance of aster microtubules in spindle orientation determination in some types of cells is well known (*Palmer et al., 1992*; *Shaw et al., 1997*), the metaphase spindles still can orient themselves parallel to the substrate in the absence of aster microtubules in both Madin-Darby Canine Kidney (MDCK) and human cervical cancer HeLa cells (*Lazaro-Dieguez et al., 2015*). In addition, the spindle orientation is controlled by multiple signaling proteins, microtubules and associated proteins, as well as actin and associated proteins (*Toyoshima and Nishida, 2007a*, *2007b*; *Woolner et al., 2008*; *Thaiparambil et al., 2012*). Although a few reports in recent years showed that the microtubule and actin cytoskeleton in interphase cells could be affected by 7–17 T ultra-high SMFs in some cell types (*Valiron et al., 2005*), information about the mitotic spindle was not provided. Therefore, although the theoretical prediction of the SMF effect on spindle orientation in cells has been experimentally tested in Xenopus embryo, how the spindles in mammalian cells, which have very different structure than Xenopus embryo, are affected is still unknown.

In our previous study, we found that prolonged treatment of 1 T moderate intensity SMF can cause multipolar spindles in cells (*Luo et al., 2016*) and we predict that stronger SMFs are likely to produce more obvious effects on spindles. To accommodate cells in the ultra-high field magnet we constructed a custom cell incubation system using two sample holders that could fit inside the new 32 mm bore ultra-high field magnets in the Chinese High Magnetic Field Laboratory (CHMFL, China). This platform provides accurate temperature and gas control for animal and human cells as well as some small model animals.

## Results

### Cells survive in 27 T SMF

The WM4 magnet we use (*Figure 1A*) provides vertical ultra-high homogenous SMF at the center of the magnet. The magnetic field direction is upward. Bright field microscopic observation and flow cytometry did not reveal obvious changes of cell morphology or cell death after 4 hr of 27 T SMF exposure in human nasopharyngeal carcinoma CNE-2Z cell (*Figure 2A–D*; *Figure 2—source data 1*). Moreover, immunostaining analysis did not reveal obvious microtubule or actin cytoskeleton abnormalities in interphase cells after 4 hr of 27 T exposure either (*Figure 2E*). Therefore 27 T SMF treatment for 4 hr is not acutely toxic to CNE-2Z cells. In addition, we found that the cell morphology was not obviously changed (*Figure 2—figure supplement 1A*) three days post-exposure but the cell number decreased by ~40% (*Figure 2—figure supplement 1B*). Flow cytometry analysis showed that the cell cycle was only slightly changed (*Figure 2—figure supplement 1C*) while the cell death was not obviously affected (*Figure 2—figure supplement 1D*).

### 27 T SMF changes spindle orientation

Tubulin and microtubules have been shown to have diamagnetic anisotropy and purified microtubules can align along the magnetic field in vitro. However, the spindle orientation after strong SMF exposure had never been experimentally investigated. Using immunofluorescence analysis of cells fixed immediately after they were taken out of the magnet, we found that mitotic spindle orientation was perturbed by 27 T SMF (*Figure 3A*). In sham incubated or control cells, the mitotic spindle axis is usually parallel to the tissue culture plate or coverslip ('lateral') while the 27 T magnetic field increased the percentage of spindles that were not parallel to the coverslip ('non-lateral') (*Figure 3A,B*). We separately measured metaphase spindles (*Figure 3C*; *Figure 3—source data 1*) as well as prometaphase and metaphase spindles together (*Figure 3—figure supplement 1*) and both results showed that non-lateral spindles were increased by 27 T SMF. We also examined lower magnetic field intensities for their effect on spindle orientation. The 0.05 T and 1 T SMFs were provided by permanent magnets placed in regular incubators (*Figure 4A*). We found that 4 hr of exposure does not increase the non-lateral spindles (*Figure 4B*; *Figure 4—source data 1*) as 27 T SMF. We also tested 9 T SMF provided by a superconducting magnet (*Zhang et al., 2016*) and found that 4 hr of 9 T exposure does not increase the non-lateral spindles either (*Figure 4C*; *Figure 4—source data 2*). However, although exposure for a prolonged time of 3 days to 0.05 T or 1 T SMFs still had no effect (*Figure 4D*; *Figure 4—source data 3*), 9 T SMF treatment for 3 days could perturb spindle

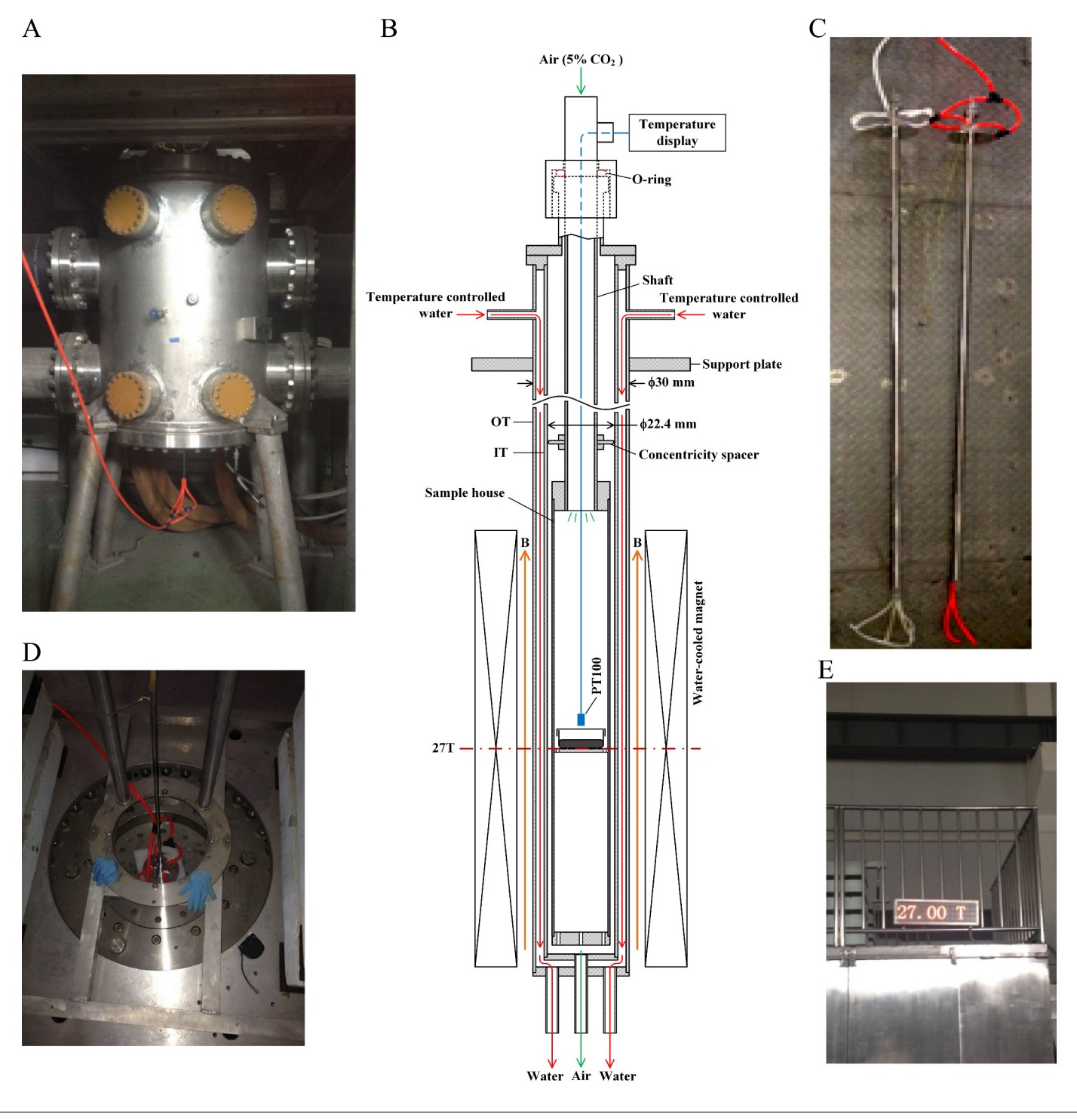

**Figure 1.** 27 T ultra-high water-cooled magnet and the biological sample incubation system. (**A**) The WM4 (water-cooled magnet#4 in the Chinese High Magnetic Field Lab). (**B**, **C**) The design and picture of the biological sample incubation system. Two identical sets were made. One was used in the magnet while the other was placed outside of the magnet to serve as the 'sham' control. (**D**) The top view of the magnet, where the biological sample incubation tube was inserted. (**E**) The magnetic field was maintained at 27 T (total of 4 hr, 3 hr stable maintenance at 27 T with half hour increase and half hour decrease).

The following figure supplement is available for figure 1:

**Figure supplement 1.** The design and actual picture of the 18 mm custom made cell culture plate.

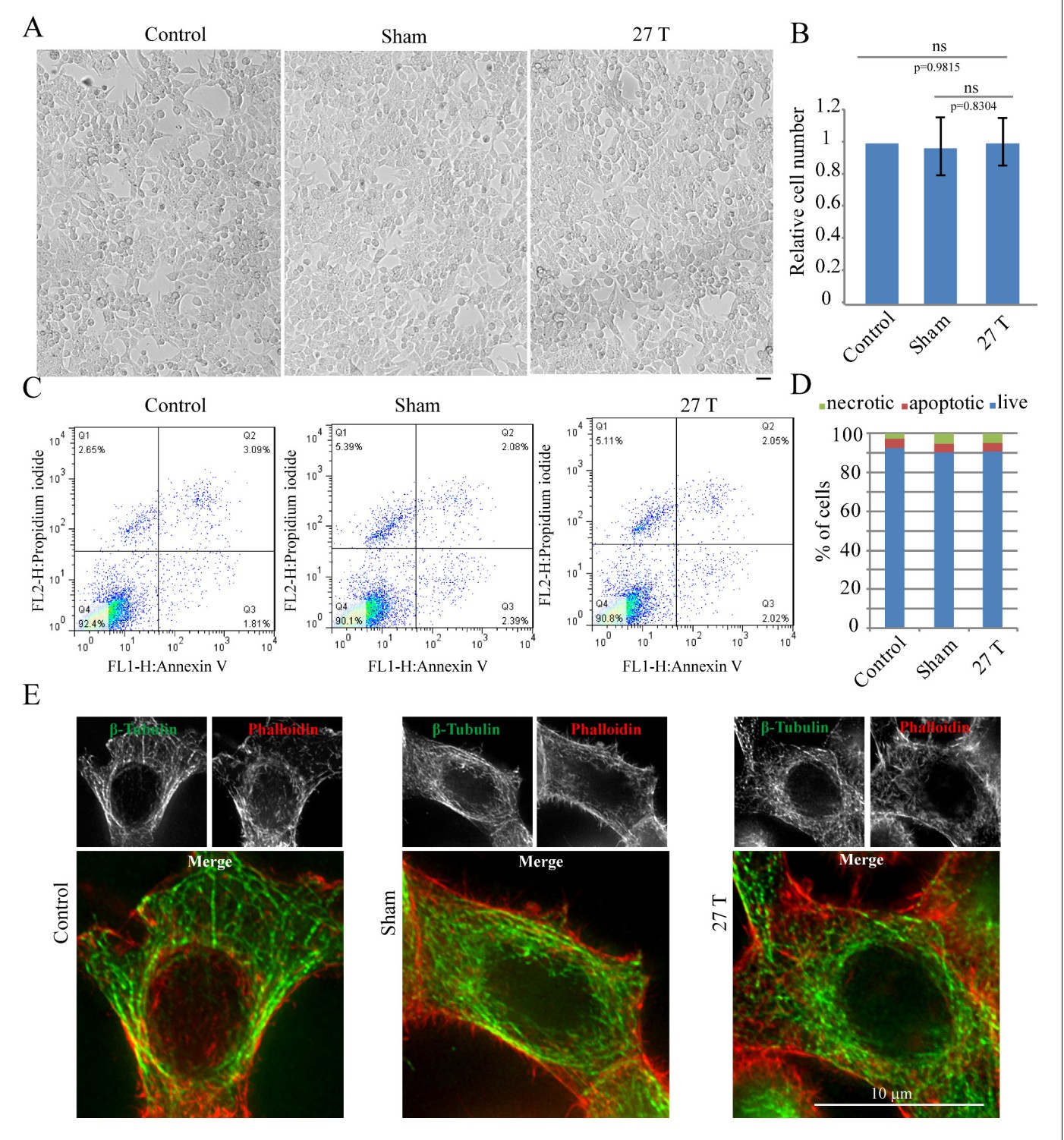

**Figure 2.** 27 T Ultra-high magnetic field does not have immediate cytotoxicity effects in CNE-2Z cells. CNE-2Z cells were plated directly on 18 mm tissue culture plate or coverslips in the18 mm tissue culture plate one night ahead to allow the cells to attach. On the day of experiment, they were placed in regular full-sized cell incubator (control) or the sample incubators in sham or in 27 T magnet for 4 hr before they were taken out and subjected to the following analysis. (**A**) Representative bright field images and of control, sham and 27 T SMF treated CNE-2Z cells. Scale bar: 20 μm. (**B**) Quantification of cell numbers in control, sham and 27 T SMF treated CNE-2Z cells from three independent experiments (n = 3). Data is mean ± SD. 'ns', not significant. (**C**) Flow cytometry results of CNE-2Z cells treated with control, sham or 27 T for 4 hr and dual staining with annexin V and PI. Bottom left leaflet shows the live cells that have intact cell membrane and have negative staining for both dyes. Top left leaflet shows necrotic cells.

*Figure 2 continued on next page*

*Figure 2 continued*

Right parts show apoptotic cells. (D) Quantification of cell numbers in each population. (E) Immunofluorescence of CNE-2Z cells shows that 27 T SMF does not have obvious effects on microtubule and actin cytoskeleton in interphase cells. CNE-2Z cells were fixed and stained with anti-tubulin antibody and fluorescently labeled phalloidin for microtubules (green) and actin (red) cytoskeleton. Experiments have been repeated for three times and representative images are shown in the figure. Scale bar: 10 μm.

The following source data and figure supplement are available for figure 2:

**Source data 1.** Quantification of cell numbers in control, sham and 27 T SMF treated CNE-2Z cells.
**Figure supplement 1.** 27 T SMF reduced CNE-2Z cell number three days post-exposure.

orientation (*Figure 4E*; *Figure 4—source data 4*). In contrast, 0.05 T and 1 T SMFs had no effect on spindle orientation even after 7 days of exposure (*Figure 4F*; *Figure 4—source data 5*). Therefore the effect of SMF on spindle orientation was field intensity-dependent and time-dependent. Ultra-high SMF of 27 T could change spindle orientation in 4 hr but 9 T SMF need 3 days to show effects.

## Spindle orientation reflects magnetic torques on chromosomes more than microtubules

Both microtubules and chromatin exhibit diamagnetic anisotropy, but prior works about spindle orientation had only assumed microtubules are the target of strong magnetic fields and did not consider the chromatin. As a start to distinguish these effects, we aligned coverslips in the same axis as the magnetic field direction, so the field should only change orientation in the plane of the coverslip (*Figure 5A–C*). We also added another non-transformed human retinal pigment epithelial cell line RPE1 to the experiment to see whether the effects we observed are cell specific (*Figure 5C*). Synchronization was also used to enrich mitotic cells (*Figure 5—figure supplement 1*). We grouped the spindles into three different groups based on their orientation relative to the field direction: parallel to the magnetic field, normal to the magnetic field and other (*Figure 5D*). Interestingly, spindle orientation was different in prometaphase cells vs. metaphase cells (*Figure 5E*). In the 27 T SMF treated group, prometaphase cells tended to orient with their spindle long axis in parallel with the field direction, while metaphase cells tended to orient with the spindle long axis normal to the field direction (*Figure 5E*). Synchronized and unsynchronized cells responded similarly (*Figure 5—figure supplement 2A,B*) so we combined data from both synchronized and unsynchronized cells for statistical analysis. As shown in *Figure 5F* (*Figure 5—source data 1*) and *Figure 5G* (*Figure 5—source data 2*), 27 T SMF had similar orientation effects on CNE-2Z and RPE1 spindles. *Figure 5—figure supplement 3* shows the Cosinus of spindle angle. The main difference between cells in prometaphase and metaphase is the organization of chromosomes, which are distributed in prometaphase and tightly aligned in metaphase. We suspect this difference accounts for the differential response to magnetic fields.

To further investigate the relative contribution of chromosomes vs. microtubules in spindle orientation, we used a CENP-E inhibitor to disrupt metaphase chromosome alignment without damaging the bipolar spindle (*Figure 6A*). In this drug, many mitotic cells exhibited highly disorganized chromosomes (*Figure 6B*). Then we measured the spindle length, chromosome distribution, metaphase plate width (*Figure 6C*) as well as the angle between spindle long axis and the magnetic field/gravity direction in Adobe Photoshop. After quantifying four independent coverslips for both CNE-2Z and RPE1 cells, we found that the chromosome distribution in spindles seemed to affect the angle between spindle long axis and the magnetic field/gravity direction (*Figure 6D*, *Figure 6—source data 1* and *Figure 6—figure supplement 1*). The preferred spindle orientation relative to the field differed between cells that have well aligned chromosomes and misaligned chromosomes, and that this difference was significant ($p < 0.01$ in RPE1 cells and $p < 0.05$ in CNE-2Z cells) (*Figure 6E*, *Figure 6—source data 2*, *Figure 6F*, *Figure 6—source data 3*). This indicates that chromosomes play a major role in spindle orientation in response to SMFs. Spindles with misaligned chromosomes tend to align with the spindle long axis in parallel with the field direction while spindles with well aligned chromosomes tend to align their metaphase plate in parallel with the field direction (*Figure 6G*).

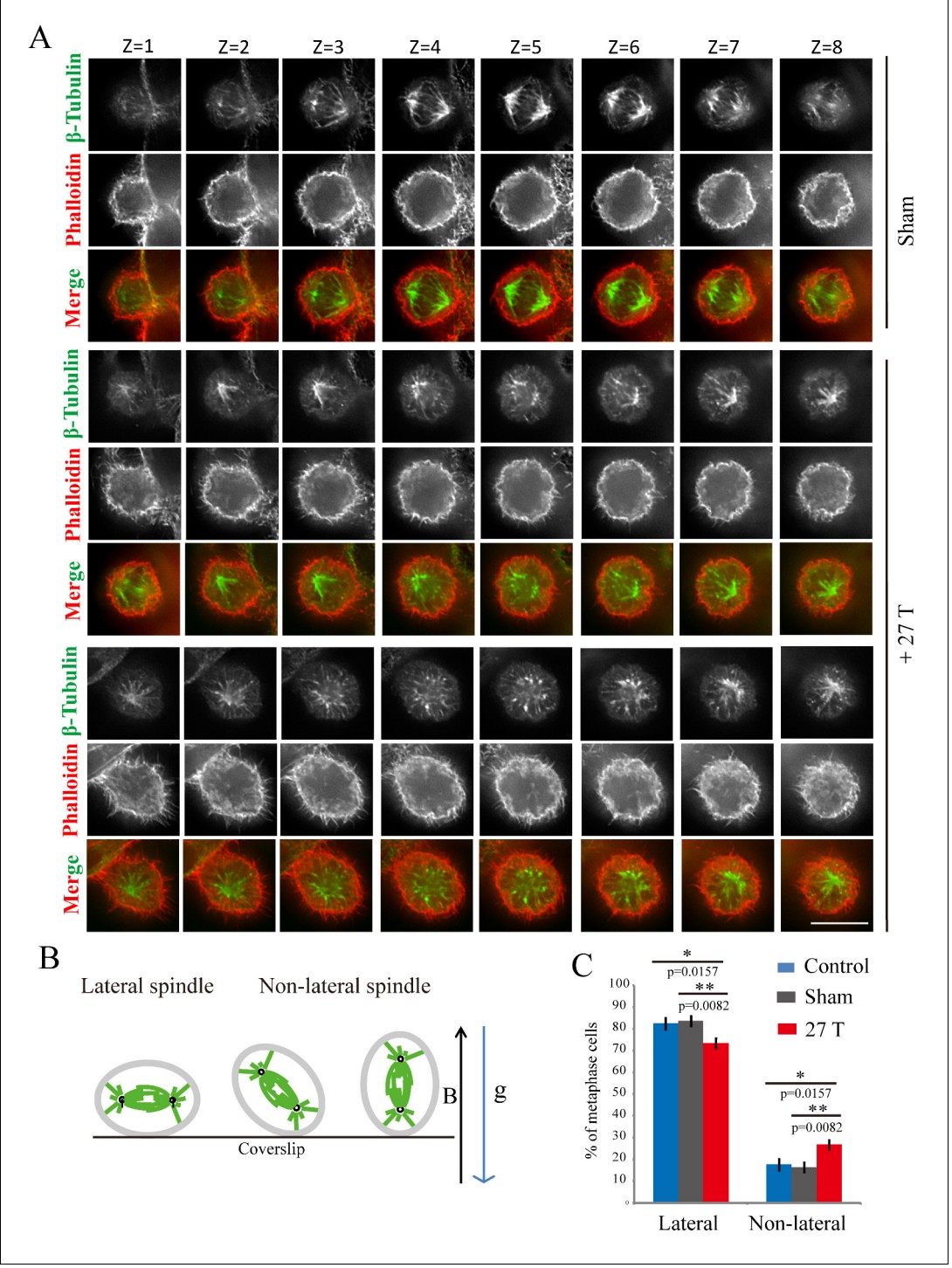

**Figure 3.** 27 T SMF changed spindle orientation. CNE-2Z cells were plated on coverslips in the 18 mm tissue culture plate one night ahead to allow the cells to attach. On the day of experiment, they were placed in regular full-sized cell incubator (control) or the sample incubators in sham or in 27 T magnet for 4 hr before they were taken out, fixed and stained with anti-tubulin antibody (for microtubules), phalloidin (for F-actin). (**A**) Representative immunofluorescence images of CNE-2Z cells show that 27 T SMF changes spindle orientation. Multistack images were taken and individual vertical image planes (Z) were displayed to show the spindle orientation. Microtubules are shown in green and F-actin filaments are shown in red. One cell in sham and two cells in 27 T SMF treated group are shown. (**B**) Illustration of spindles with different orientations. 'B' shows the magnetic field direction and 'g' shows the gravity direction. (**C**) Quantification of spindle orientations in control,

*Figure 3 continued on next page*

*Figure 3 continued*

sham or 27 T treated cells from four independent experiments (n = 4). Data is presented as mean ± SD. *p<0.05; **p<0.01. Total of 921 metaphase spindles were counted.

The following source data and figure supplement are available for figure 3:

**Source data 1.** Quantification of spindle orientations in control, sham or 27 T treated cells from four independent experiments.

**Figure supplement 1.** Spindle orientation in prometaphase and metaphase CNE-2Z cells were changed by 27 T SMF.

Not surprisingly, lower magnetic fields of 1 T or 9 T treatment for 4 hr did not affect the spindle orientation (*Figure 6—figure supplement 2*).

## Magnetic torque on spindles affects spindle morphology

We also noted that spindles aligned normal to the magnetic field direction were wider than those parallel to the field direction (*Figure 7A*). We carefully compared it in control and 27 T SMF treated cells (*Figure 7B*). In both synchronized and unsynchronized CNE-2Z and RPE1 cells, the spindles in 27 T SMF-treated cells that were normal to the field direction has bigger spindle pole angles, which makes them look 'wider' (*Figure 7C*, *Figure 7—source data 1* and *Figure 7—figure supplement 1*) compare to the ones parallel to the field direction. In control cells that were not treated with SMFs, the spindle pole angles are similar in both directions (*Figure 7C*). To get a more quantitative measurement, we quantified the spindle length (a) and width (w) and found that the spindle length was not much affected by the 27 T magnetic field (*Figure 7—figure supplement 2A,B*) but the spindle width was increased in both CNE-2Z and RPE1 cells, when the spindle was normal to the field direction (*Figure 7D*, *Figure 7—source data 2* and *Figure 7—figure supplement 2C,D*), which confirms that the spindle is 'wider'.

To find out whether chromosome alignment affected the spindle morphology change in the presence of 27 T field, we analyzed the spindle dimension and chromosome distribution (*Figure 7E*) for CENP-E inhibitor treated spindles that are normal to the field direction (the angle between spindle long axis to the magnetic field direction within 80–90 degree). It is interesting that the chromosome alignment or misalignment did actually affect the spindle ellipticity (*Figure 7F*, *Figure 7—source data 3*). Our results show that spindles with well aligned chromosomes became obviously wider in 27 T (p<0.005), but the morphology of spindles with misaligned chromosomes was not much affected (*Figure 7F*). We also noticed that spindles with well aligned chromosomes themselves are slightly wider in the sham control group compared to spindles with misaligned chromosomes (*Figure 7F*), which may have their microtubules relative easier to be aligned by a vertical magnetic field.

## Discussion

We propose that the SMF-induced spindle orientation and morphology changes are due to the combined alignment effects of both microtubules and chromosomes in the magnetic field (*Figure 8*). When the field is oriented normal to the substrate, magnetic torques on both microtubules and chromatin may combine to re-orient spindles away from the surface plane, opposing torques on astral microtubules that promote the normal orientation. Application of the magnetic field parallel to the coverslip allowed us to discriminate torques on chromatin vs. microtubules, and in this case it appears that torques on well aligned chromatin dominated, aligning spindles preferentially with their microtubules normal to the field, and their metaphase plate parallel to the field (*Figures 5* and *6*). Our result is opposite to that proposed by Denegre et al, who studied Xenopus egg cleavage in 16.7 T high magnetic fields and theoretically proposed that the orientation of spindle in magnetic field is a result of the balance between aster microtubules and spindle microtubules (*Denegre et al., 1998*). They did not directly image spindles, but we note opposite results in the two systems might depend on huge size-scaling differences between the two systems we used. The cleavage plane in

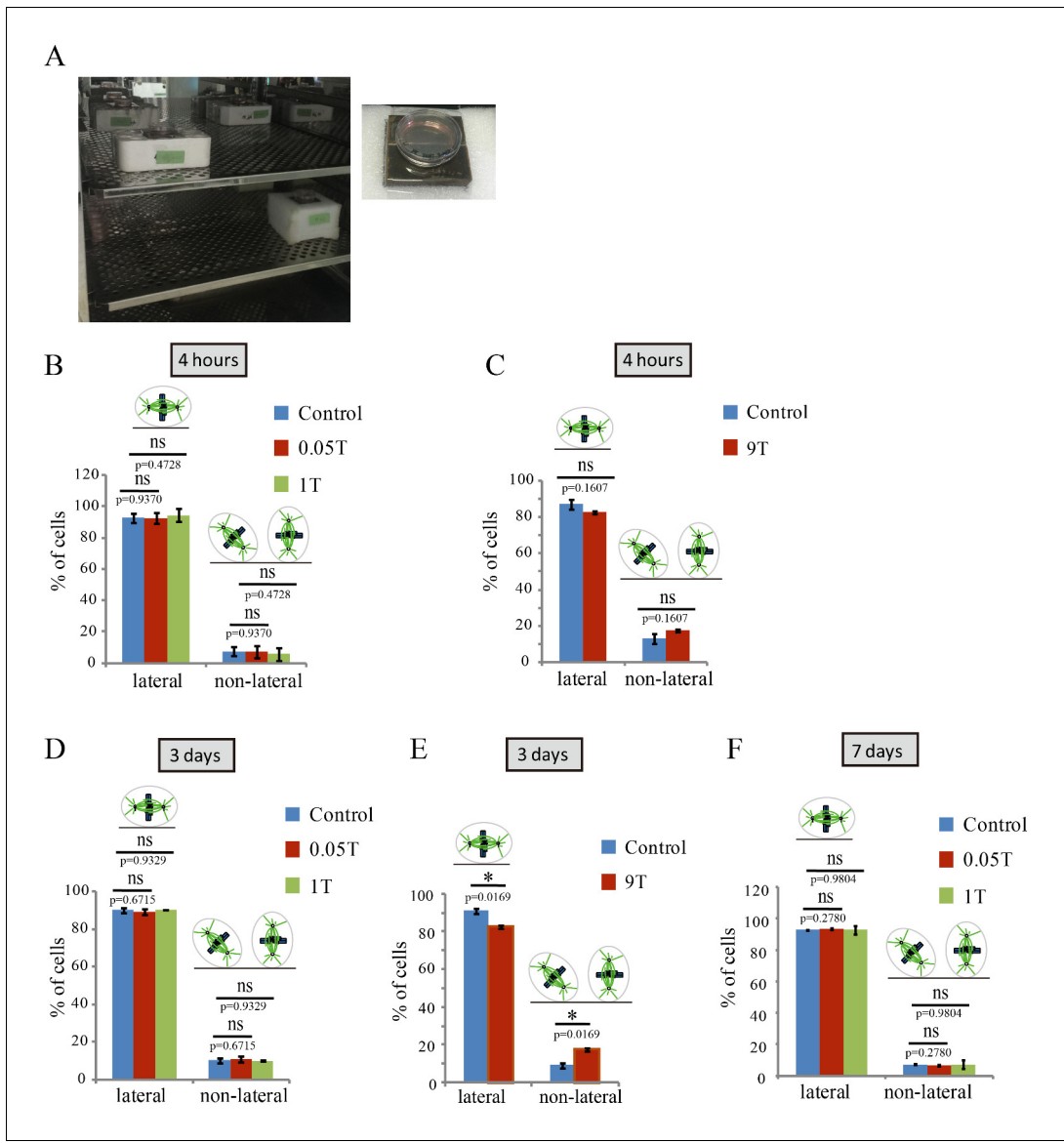

**Figure 4.** SMFs affected spindle orientation in a field intensity dependent manner. CNE-2Z cells were plated on coverslips in the 35 mm or 24 well tissue culture plate one night ahead to allow the cells to attach. On the day of experiment, they were exposed to different intensity SMFs for different time before they were taken out, fixed and stained with anti-tubulin antibody (for microtubules) and DAPI (for DNA). (**A**) 0.05 T and 1 T moderate intensity SMF exposure experimental set-up. Permanent magnets were placed inside a regular full-sized cell incubator to ensure proper culture conditions. Cell culture plate was placed on the top surface center of the magnet. (**B–F**) Quantification of spindle orientations in control, 0.05 T, 1 T or 9 T treated cells. Total of 200–300 metaphase spindles from 3–4 independent coverslips were examined for each condition. Data is presented as mean ± SD. 'ns', not significant; *p<0.05.

The following source data is available for figure 4:

**Source data 1.** Quantification of spindle orientations in control, 0.05 T or 1 T treated cells (4 hr treatment).
**Source data 2.** Quantification of spindle orientations in control, or 9 T treated cells (4 hr treatment).
**Source data 3.** Quantification of spindle orientations in control, 0.05 T or 1 T treated cells (3d treatment).
**Source data 4.** Quantification of spindle orientations in control, or 9 T treated cells (3d treatment).

*Figure 4 continued on next page*

*Figure 4 continued*

**Source data 5.** Quantification of spindle orientations in control, 0.05 T or 1 T treated cells (7d treatment).

Xenopus eggs is oriented by huge microtubule asters with almost mm dimensions, while the amount of chromatin is the same as a mitotic cell. In contrast, the ratio of chromatin to microtubules is much larger in human somatic cells. Theoretical calculation predicted that the highly compacted mitotic chromosomes could be fully aligned by magnetic fields at around 1.4 T (*Maret, 1990*). Therefore the metaphase plate composed of chromosomes likely dominated the SMF-induced orientation in normal-sized somatic cells.

This is the first reported study that investigated the mammalian cellular responses to ultra-high magnetic field of above 20 T. The torque of a substance is equal to the product of magnetic field intensity and the magnetic susceptibility of the object, in which the susceptibility could also be field-dependent and can be Taylor expanded. This means that the torque could be parabolically proportional to the magnetic field strength. Thus, a high field has much more severe impact than a low field as far as field-induced alignment is concerned, and the relationship is not linear. Although there are various potential concerns for the safety issue of the high magnetic fields, the known biological effects of high fields of above 10 T are limited. There are only a few studies that have investigated the animal or human cells at $\geq$10 T. Nakahara et al and Zhao et al showed that 10 T or 13 T SMFs did not have obvious effects on the human-hamster hybrid (AL) cells, Chinese Hamster Ovary (CHO) cells or human primary skin fibroblasts (AG1522) cells (*Nakahara et al., 2002*; *Zhao et al., 2010*); Denegre et al found that 16.7 T large gradient SMFs can affect the division of Xenopus eggs, which is presumed to act through astral microtubules and spindles (*Denegre et al., 1998*); the Shang group found that 12-16-12T large gradient SMFs can affect microtubule actin crosslinking factor 1 (MACF1) expression and its association with cytoskeleton (*Qian et al., 2009*) as well as osteoblast ultrastructure and cell viability by disrupting collagen I or Febronectin/$\alpha\beta$1 integrin in human osteoblast-like cell lines (*Qian et al., 2013*); Valiron et al used 17 T, the highest SMF strength applied on cells so far and they did not observe strong cell killing effects (*Valiron et al., 2005*). Instead, they found that the cytoskeleton of interphase mouse embryo fibroblast 3T3 cells and human cervical cancer HeLa cells can be affected by 17 T SMF. Here we found that 27 T SMF does not have an immediate cell killing effects on human CNE-2Z and RPE1 cells but it changes the mitotic spindle orientation and morphology.

The spindle orientation change induced by ultra-high SMF is likely conserved between different cell types, although there may be variations among them. The spindles align with their long axis vertical to the field direction and their metaphase plate parallel to the field direction are different from our prediction. We thought the microtubules within the spindle would align with the magnetic field direction so that the spindle would also align with the field. However, it has been shown that the mitotic chromosomes have electromagnetic properties (*Zhao and Zhan, 2012*). Although the chromosomes alignment in the magnetic field has not been experimentally shown, calculation predicted that the chromosomes could be fully aligned with the magnetic field as long as the field is above 1.4 T (*Maret, 1990*). Based on theoretical calculation and experimental validation, Valles and his colleagues proposed that the direction of mitotic spindle alignment in the magnetic field depends on how the diamagnetic anisotropies of its individual components, including microtubules and chromosomes, align within it (*Valles, 2002*; *Valles et al., 2002*). Denegre et al's results about Xenopus egg division shows that the cleavage plane, which is perpendicular to the spindle long axis, tend to parallel with the magnetic field direction (*Denegre et al., 1998*). Xenopus eggs have very different long astral microtubules, which composed a large portion of the whole cell. In contrast, the cells used in our study as well as most human somatic cells have much less astral microtubules and much bigger spindle microtubules compared to Xenopus eggs. Since both astral and spindle microtubules, as well as chromosomes respond to magnetic field, their relative contribution will be critical to determine the final outcome in a given cell type.

In conclusion, although it is well accepted that microtubules can be affected by magnetic fields, the effect of high magnetic field on mitotic spindles in a cell has never been investigated. Here we examined the mitotic spindles in 27 T ultra-high SMF treated human CNE-2Z and RPE1 cells and found that the spindle orientation is indeed affected and the direction is dependent on the

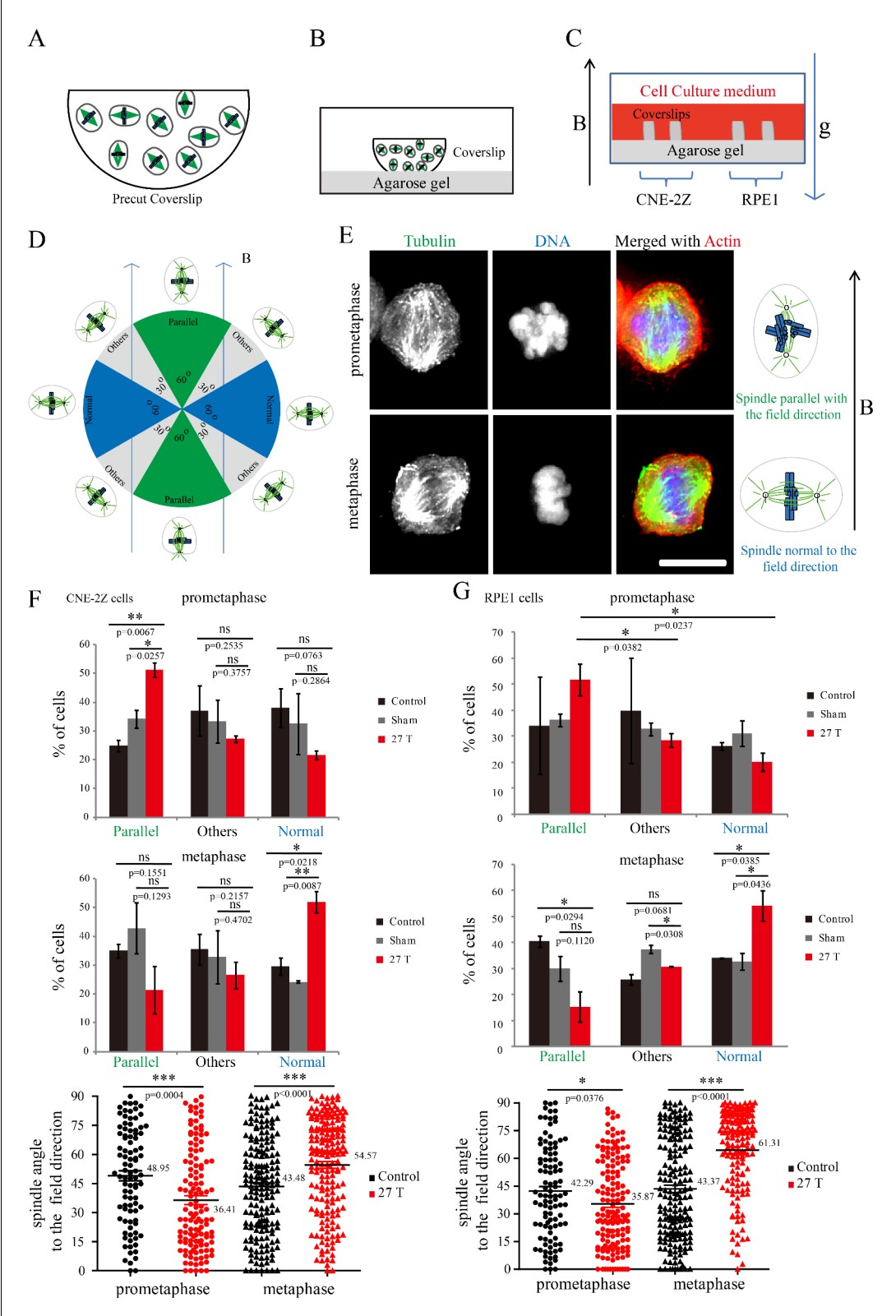

**Figure 5.** Prometaphase and metaphase spindles have different orientations in 27 T SMFs. (A–C) Schematic illustration of the experimental set-up. (A) CNE-2Z and RPE1 cells were plated on pre-cut coverslips one night ahead to allow the cells to attach. (B) On the day of experiment, the coverslips were inserted onto agarose gel in the 18 mm plates. (C) Cells were treated with or without synchronization, and with or without 27 T magnetic field for 4 hr before they were fixed and stained with anti-tubulin antibody (for microtubules) and fluorescently labeled phalloidin (for actin polymer) and DAPI (for

*Figure 5 continued on next page*

*Figure 5 continued*

DNA). 'B' shows the magnetic field direction and 'g' shows the gravity direction. (D) The orientation of the spindle long axis was measured and characterized into 'parallel' (green), 'normal' (blue) and 'others' (grey) according to the angle between spindle long axis and the magnetic field direction. (E) Representative immunofluorescence images of prometaphase and metaphase RPE1 cells that have different orientation when they were exposed to 27 T SMF for 4 hr. Scale bar: 10 μm. (F, G) Quantification of prometaphase and metaphase spindle orientations in control, sham or 27 T treated CNE-2Z (F), and RPE1 (G) cells. One experiment was done in synchronized cells and the other was done with unsynchronized. Total of 1575 spindles were measured from four independent coverslips from two independent experiments. The histograms were created in excel (mean ± SD). Scatter plots were created in GraphPad (mean ± SEM). *p<0.05, **p<0.01, ***p<0.005.

The following source data and figure supplements are available for figure 5:

**Source data 1.** Quantification of prometaphase and metaphase spindle orientations in control, sham or 27 T treated CNE-2Z cells.

**Source data 2.** Quantification of prometaphase and metaphase spindle orientations in control, sham or 27 T treated RPE1 cells.

**Figure supplement 1.** Synchronization procedure to enrich mitotic cells and spindle orientation measurement.

**Figure supplement 2.** Prometaphase and metaphase cells show different orientation in both synchronized and unsynchronized CNE-2Z and RPE1 cells after 27 T SMF exposure for 4 hr.

**Figure supplement 3.** Cosinus of angles between spindle long axis and the 27 T magnetic field direction.

chromosome alignment. This provides a powerful tool to study spindle orientation related questions in developmental biology, such as cell fate, tissue architecture, and cancer biology.

## Materials and methods

### Materials

The 18 mm cell culture plates were custom made by Guangzhou Jet Bio-Filtration Company, China. The antibody for $\beta$-tubulin (#HC101) was from Beijing TransGen Biotech. The Alexa Fluor 488 and 594 Phalloidin (RRID: AB_2315147 and RRID: AB_2315633), secondary antibodies for immunofluorescence and anti-fade ProLong Gold with DAPI were all from Invitrogen (RRID: SCR_008410). The 1 T magnets (Neodymium magnet, N38, dimension of 5 cm x 5 cm x 5 cm) were from China Dafeng Zhongxin Permanent Magnet Material. The CENP-E inhibitor (GSK923295, # S7090) was from Selleck. FITC-Annexin V Apoptosis Detection Kit was from BD Pharmingen.

### Construction of the biological sample incubation system for 27 T ultra-high magnet

The 27 T water-cooled magnet (WM) in Chinese High Magnetic Field Laboratory (CHMFL, China) facility (WM#4) has a 32 mm diameter room temperature bore. *Figure 1* shows the device that can fit the 27 T WM. The device consists of two coaxial non-magnetic stainless steel tubes. The outer diameter of the outer tube (OT) is 30 mm and the inner diameter of the inner tube (IT) is 22.4 mm. To investigate the biological effect of the magnetic field provided by The WM4 (*Figure 1A*), we designed (*Figure 1B*) and constructed a set of biological sample incubation system (*Figure 1C*, and *Figure 1—figure supplement 1*), with accurate temperature, gas and humidity control. A non-magnetic stainless steel tube with 10 mm outer diameter was used as a shaft and was inserted into the inner space of the IT. We have used a rubber O-ring to hold the shaft and sealed the inner space of the IT. A sample house and a Teflon concentricity spacer were fixed on the shaft. The concentricity spacer was used to make sure that the shaft and the sample house are coaxial with the IT. Because the shaft was held by the rubber O-ring, it can be moved on axial easily to adjust the position of the sample house in the WM. A PT100 near the samples was used as a temperature sensor and connected to a temperature display to monitor the temperature of the samples. The temperature of the samples can be controlled by thermal conduction from the temperature controlled water, which flow through the space between the IT and OT. By adjusting the temperature of the water, the

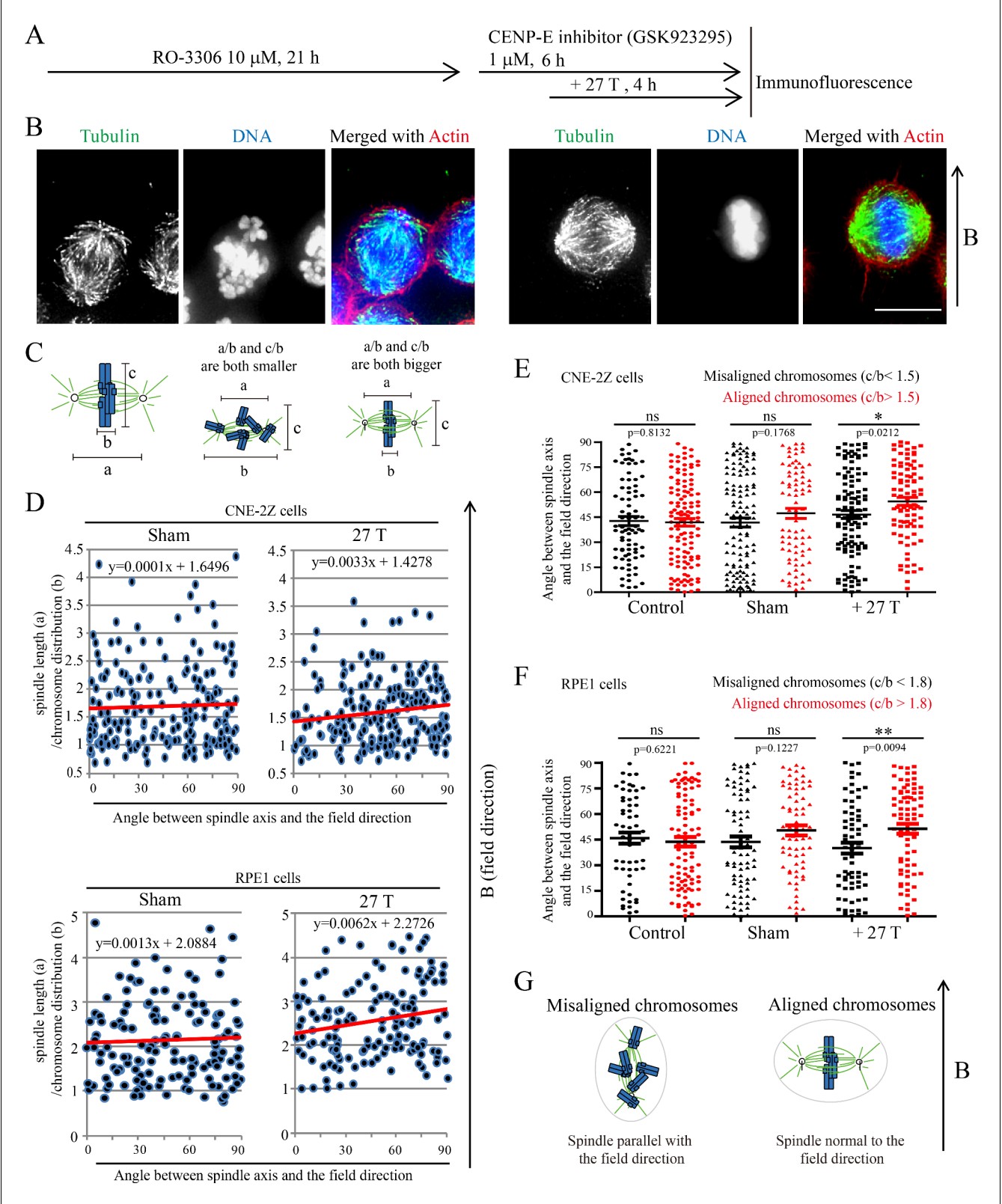

**Figure 6.** Chromosome alignment affects spindle orientation. CNE-2Z and RPE1 cells were treated with RO-3306 and the CENP-E inhibitor (GSK923295) before they were exposed to 27 T SMF for 4 hr. Cells were then harvested for Immunofluorescence experiment. (**A**) Schematic illustration of the experimental procedure. (**B**) Representative immunofluorescence images show that CNE-2Z cells with misaligned chromosomes prefer to align with their spindle long axis in parallel with the magnetic field direction (upward) but the cells with well aligned chromosomes prefer to align their

*Figure 6 continued on next page*

*Figure 6 continued*

chromosome plate in parallel, with the spindle long axis normal to, the magnetic field direction. Scale bar: 10 µm. (C) Schematic illustration of the spindle and chromosome measurement. 'a' is the spindle length, defined as the distance between two spindle poles. 'b' is the chromosome distribution, defined by the maximum distance between chromosomes along the spindle long axis. 'c' is the metaphase plate width. Measurements were done in Adobe Photoshop. (D) The 27 T SMF affects the angle between spindle long axis with magnetic field direction in CNE-2Z and RPE1 cells, which is determined by chromosome distribution. Sham and 27 T groups are shown in the figure. Control groups are shown in the figure supplement due to space limitation. (E, F) Quantification of the angle between spindle long axis with the magnetic field direction in CNE-2Z (E) or RPE1 (F) cells in control, sham control or 27 T SMF treated group to compare the difference between spindles with misaligned vs. aligned chromosomes. 'Misaligned chromosomes' were defined as c/b <1.5 (CNE-2Z cells) or 1.8 (RPE1 cells) and 'Aligned chromosomes' were defined as c/b >1.5 (CNE-2Z cells) or 1.8 (RPE1 cells). Quantifications for D-F were from total of 618 CNE-2Z spindles and 452 RPE1 spindles. Spindles for each cell type were from four independent coverslips in two independent days. Data is mean ± SEM. 'ns', not significant; *$p<0.05$; **$p<0.01$. (G) Cartoon illustrates that spindles with misaligned chromosomes tend to align along the magnetic field direction (B, upward) while spindles with compact metaphase plate tend to align normal to the field direction.

The following source data and figure supplements are available for figure 6:

**Source data 1.** The 27 T SMF affects the angle between spindle long axis with magnetic field direction in CNE-2Z and RPE1 cells, which is determined by chromosome distribution.
**Source data 2.** Quantification of the angle between spindle long axis with the magnetic field direction in CNE-2Z cells in control, sham control or 27 T SMF treated group to compare the difference between spindles with misaligned vs. aligned chromosomes.
**Source data 3.** Quantification of the angle between spindle long axis with the magnetic field direction in RPE1 cells in control, sham control or 27 T SMF treated group to compare the difference between spindles with misaligned vs. aligned chromosomes.
**Figure supplement 1.** The chromosome distribution and angle between spindle long axis and the magnetic field/gravity direction in control CNE-2Z and RPE1 cells.
**Figure supplement 2.** SMFs of 1 T and 9 T did not affect spindle orientation in CNE-2Z cells.

temperature of the samples can be controlled precisely. To adjust the atmosphere of the sample house, the air with 5% $CO_2$ with temperature and humidity control was introduced by the shaft.

To control for possible effect of the incubation system we made two identical sets, one for 'sham' and the other for the experimental 27 T SMF exposure group (*Figure 1C*). The experimental group was placed in the tube and inserted into the WM4 (*Figure 1D*), while the sham group was placed in the other identical tube and left outside of the magnet. The control group was kept in the regular full-size $CO_2$ cell incubator in the lab. Although the maximal magnetic field intensity that WM4 can reach is 27.5 T, here we used 27 T to ensure the stability during the whole experiment process (*Figure 1E*).

The biological sample incubation system we constructed can hold samples up to 18 mm in diameter and incubate at temperature range of 4–100 degree. We custom made small plates to fit in this biological sample incubation system. Most experiments in this study were carried out in these custom made 18 mm plates, including full size incubator control, sham control and the 27 T experimental groups. Since the gas, humidity and temperature can all be well controlled, the platform we built is suitable to study a wide range of biological samples, such as various cell cultures, small model animals such as fruit flies, C elegans, zebrafish and mouse tissues.

## Cell culture

CNE-2Z cells (RRID:CVCL_6890) were cultured in RPMI-1640 (#10–040-CVR, CORNING Life Sciences) supplemented with 10% FBS and 1% P/S (penicillin/streptomycin). RPE1 cells (RRID: CVCL_4388) were maintained in DMEM (#10–017-CV, CORNING Life Sciences) supplemented with 10% FBS and 1% P/S. Both cells were maintained in cell incubator with 5% $CO_2$, at 37°C. Both cell lines were from ATCC and have been confirmed by STR profiling. Mycoplasma has been tested to be negative.

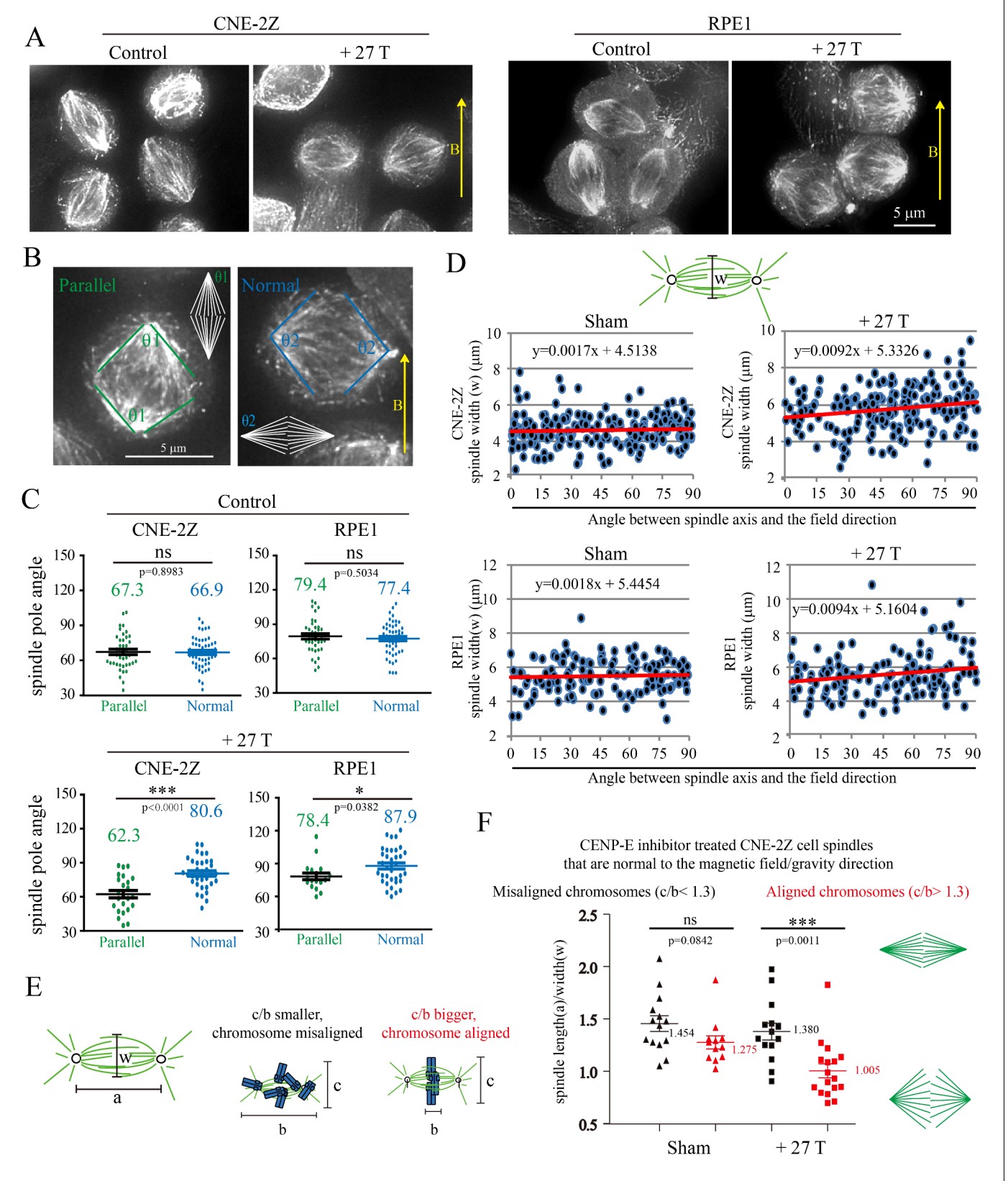

**Figure 7.** 27 T SMF changes spindle morphology in both CNE-2Z and RPE1 cells. (**A**) Representative immunofluorescence images of CNE-2Z and RPE1 cells with or without 27 T SMF treatment for 4 hr. Scale bar: 5 μm. (**B**) Illustration of the pole angle measurement of the metaphase spindles in CNE-2Z

*Figure 7 continued on next page*

*Figure 7 continued*

and RPE1 cells with or without 27 T SMF treatment for 4 hr. '1' measures the pole angle of metaphase spindles in parallel to the magnetic field/gravity direction (green) and '2' measures the pole angle of metaphase spindles normal to the magnetic field/gravity direction (blue). Scale bar: 5 µm. (C) Quantification of the metaphase spindle pole angle measurement for synchronized CNE-2Z and RPE1 cells with or without 27 T SMF. RO-3306 and MG132 synchronization was used to increase the percentage of mitotic cells. Total of 295 metaphase spindles were measured from four independent coverslips. Data is mean ± SEM. (D) Quantification of the spindle width for RO-3306 and CENP-E inhibitor treated CNE-2Z and RPE1 cells. Experimental procedure was as shown in *Figure 5A*. Sham and 27 T treated groups are shown here and the control groups are shown in *Figure 7—figure supplement 2C,D*. (E) Illustration of the spindle and chromosome measurement. (F) Quantification of the relationship between spindle morphology and chromosome alignment in CENP-E inhibitor treated CNE-2Z cells that have spindle axis normal to the magnetic field/gravity direction (angle of 80–90 degree). Misaligned chromosomes (black) vs. aligned chromosomes (red) were classified by different c/b ratio values. Measurement was done on spindles from four independent coverslips from two independent experiments. Data is mean ± SEM. 'ns', not significant; *p<0.05; ***p<0.005.

The following source data and figure supplements are available for figure 7:

**Source data 1.** Quantification of the metaphase spindle pole angle measurement for synchronized CNE-2Z and RPE1 cells with or without 27 T SMF.
**Source data 2.** Quantification of the spindle width for RO-3306 and CENP-E inhibitor treated CNE-2Z and RPE1 cells.
**Source data 3.** Quantification of the relationship between spindle morphology and chromosome alignment in CENP-E inhibitor treated CNE-2Z cells that have spindle axis normal to the magnetic field/gravity direction (angle of 80–90 degree).
**Figure supplement 1.** Quantification of the spindle pole angle measurement in unsynchronized CNE-2Z and RPE1 cells.
**Figure supplement 2.** Spindle length was not affected by 27 T SMF when spindle axis was normal to the field direction.

## Immunofluorescence

CNE-2Z and RPE1 cells were placed on coverslips in 18 mm, 24-well plates or 35 mm cell culture plates and treated with 27 T, 9 T, 1 T or 0.05 T magnetic fields. Cells were washed once with PBS and fixed by 4% (vol/vol) formaldehyde at room temperature for 20 min. Then the coverslips were washed with TBS-Tx (TBS supplemented with 0.1% Triton X-100) and blocked by AbDil-Tx (TBS-Tx with 2% (wt/vol) BSA and 0.05% sodium azide) at room temperature for at least 30 min. Coverslips were stained with anti-$\beta$-tubulin antibody at room temperature for 2 hr, followed by fluorescently conjugated secondary antibodies at room temperature for 1 hr. Then the cells were directly stained with fluorescently labeled Phalloidin for 1 hr at room temperature. After washing with TBS-Tx, coverslips were mounted in anti-fade ProLong Gold mounting medium with DAPI (Invitrogen). The antibodies and reagents used in immunofluorescence experiments include $\beta$-tubulin (used at 1:1000 dilution), the secondary fluorescently conjugated antibodies (used at 1:250 dilution), as well as Alexa 488 or 594 Phalloidin (used at 1:40 dilution).

## Cell counting and cell cycle analysis

All attached and floating cells were collected for cell counting, apoptosis and cell cycle analysis. Bright field images were taken before the cells were harvested by trypsinization. An aliquot of the cells were counted by hemocytometer and the rest cells were used for flow cytometry analysis (cell death and cell cycle). Most experiments were repeated for at least three independent times by two researchers. The results were gathered together for analysis.

Cells were trypsinized and washed three times with PBS before they were fixed in 70% ice-cold ethanol overnight at 4°C. Then they were washed with PBS again, and incubated in PI (propidium iodide) solution (BD Pharmingen) for 30 min in the dark at room temperature. Samples were then analyzed on a BD Flow Cytometry (RRID: SCR_013311) (BD bioscience, Calibur). $1 \times 10^4$ cells per sample were collected for each condition. Data were analyzed by ModFit LT. Experiments were done for at least three times.

## Apoptosis assay

Cells were trypsinized and washed twice with ice-cold PBS before they were resuspended in binding buffer at $1 \times 10^6$ cells/ml. Then 100 µl of them was transferred to a 1.5 ml culture tube. FITC-

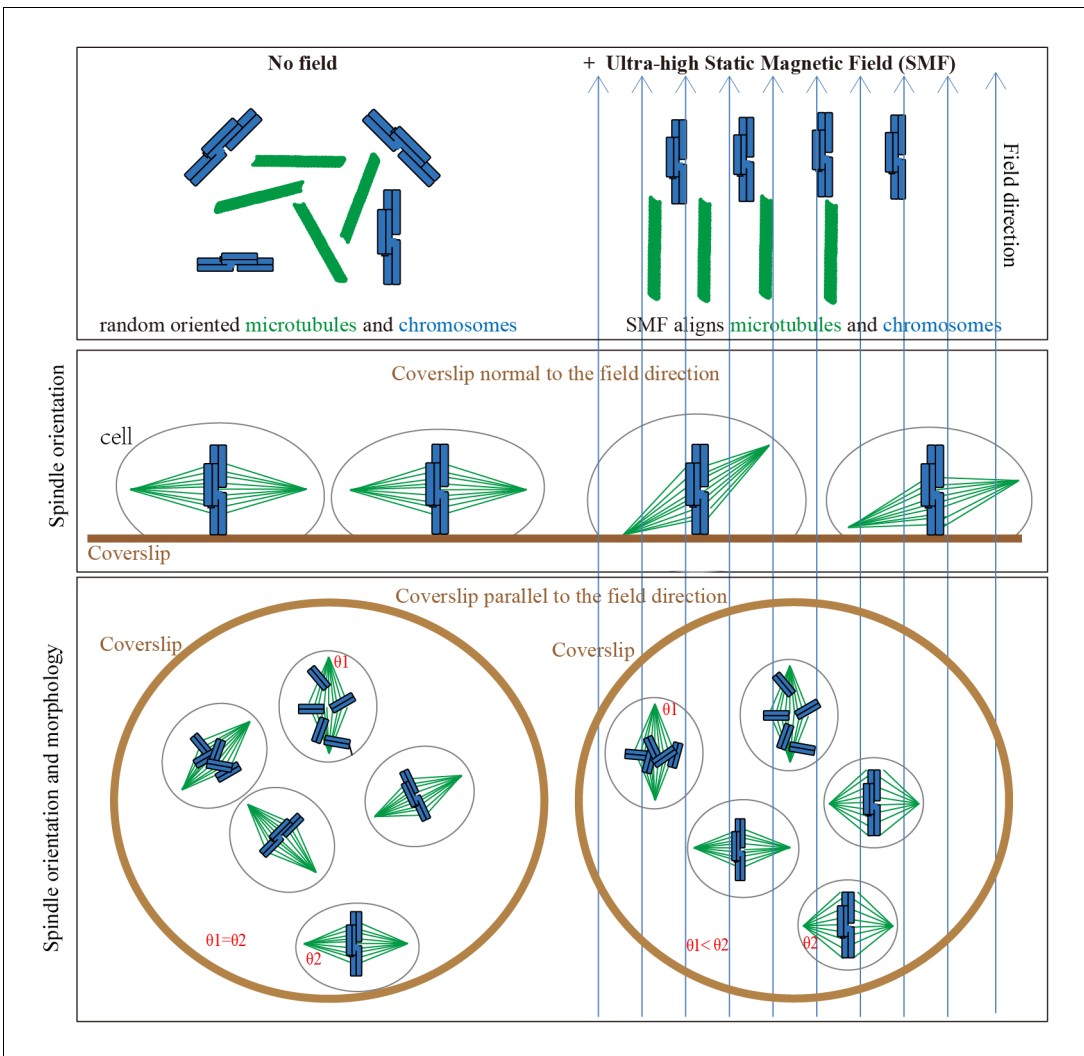

**Figure 8.** Models show that ultra-high SMFs align microtubules and chromosomes to change spindle orientation and morphology. Blue upward arrows show magnetic field direction. Cells were plated on coverslips, which were placed in the ultra-high magnetic field either normal to or in parallel with the field direction.

Annexin V (5 µl) and PI (5 µl) were added to the tube, mixed, and incubated for 15 min at room temperature in the dark. Then 400 µl of binding buffer was added to the stained cells and mixed before they were analyzed by flow cytometry within 1 hr. Approximately $1 \times 10^4$ cells were collected by flow cytometer. Data were analyzed by FlowJo. Experiments were done for at least three times and representative results were shown in the figures.

## Microscopy

The bright field images were taken by a DSZ2000 microscope equipped with ISH300 3.0MP camera (UOP). Most Immunofluorescence images shown in the figures were using a DeltaVision microscope (GE Healthcare) equipped with a 60× objective lens, 0.5 µm step size. Deconvolved images were projected into a single picture using Image J software (RRID:SCR_003070) and maximum projection images are shown in the figures. Some low magnifications immunofluorescence images were taken by a Leica DMI4000B microscope (RRID:SCR_000011) for spindle measurement.

## Magnetic field exposure

Since most high field MRIs for preclinical or research uses are applied in the head region, including the highest field MRI (21.1 T) used in mice (*Schepkin et al., 2010*), we chose a human nasopharyngeal carcinoma CNE-2Z cell line for its potential clinical relevance. Cells were plated on round or pre-cut coverslips one night before to allow the cells to attach. They were then exposed to different magnetic fields for indicated time points and then removed for analysis. Bright field microscopic and annexin V/PI double stain in flow cytometry assays were used to analyze apoptosis and necrosis

For 0.05 T and 1 T magnetic field exposure perpendicular to coverslips, CNE-2Z cells on coverslips were exposed to magnetic fields for 4 hr, 3 days or 7 days in regular full-sized $CO_2$ cell incubator (Shanghai Boxun, BC-J160S) that has accurate control of temperature (37°C), humidity and $CO_2$ (5%). The cell plates were placed right on the top center of the magnets (Neodymium magnet N38, dimension: 5 cm x 5 cm x 5 cm) and the magnetic field intensity measured by a Gauss meter (Lake-Shore 475 DSP Gaussmeter) showed the magnetic field intensity of 1.07 T (10700 Gs) for the 1 T magnet. The sham control group was in the same incubator, around 30–40 cm away from the magnets, where the magnetic field is 0.925 Gs (background magnetic field in the lab was measured to be $0.875 \pm 0.171$ Gs and in a separate $CO_2$ cell incubator with no magnets was $0.875 \pm 0.096$ Gs).

For 9 T magnetic field (upward direction) exposure perpendicular to coverslips, 2 ml of $5 \times 10^5$ cells /ml CNE-2Z cells for 4 hr assays, or 2 ml of $1.25 \times 10^5$ cells /ml CNE-2Z cells for 3 days assays were placed on coverslips in each 35 mm cell culture plates one night ahead to allow the cells to attach. On the second day, they were placed in 5% $CO_2$ at 37°C under 9 T SMF for 4 hr (3 hr stable maintenance at 9 T with half hour increase and half hour decrease) or 3 days before they were fixed and stained for analysis.

For 27 T magnetic field exposure perpendicular to coverslips, 400 µl $6.25 \times 10^5$/ml CNE-2Z cells were placed on coverslips in each 18 mm cell culture plates one night ahead to allow the cells to attach. On the second day, they were placed in regular full-sized cell incubator (control group) or in two high magnetic field biological sample incubation systems. One was used as sham group and the other was 27 T group. Then the 27 T group was placed in the center of the 27 T water-cooled magnet (WM4) and the whole cell culture plate was exposed homogeneously to the 27 T magnetic field for 4 hr in total (increasing field for 30 min, constant 27 T field for 3 hr and reducing field for 30 min). For the 3 days post-exposure experiments, cells taken out of the magnet were then maintained in regular full-sized cell incubator for another 3 days before they were taken out and subjected to further analyses. The sham group was processed identically.

For 1 T, 9 T or 27 T magnetic field exposure parallel to coverslips, coverslips were cut in half using a glass cutter and $1 \times 10^7$ CNE-2Z or RPE1 cells were plated in 100 mm plate one night ahead to allow the cells to attach. On the second day, the half coverslips were inserted vertically into the agarose gel pre-plated and solidified on the bottom of 35 mm (1 T and 9 T groups) or 18 mm (27 T group) plates. 1 ml (1 T and 9 T groups) or 400 µl (27 T group) of 1.5% agarose was used for each plate. They were placed in regular full-sized cell incubator (control group), in 1 T, 9 T or in two high magnetic field biological sample incubation systems. One was sham group and the other was 27 T group. The rest of the experiment was identical to the above mentioned experiments.

## Spindle measurement

To measure the spindle angles with the coverslips (lateral or non-lateral) in *Figures 3–4*, we used the Leica DMI4000B fluorescent microscope and observed the spindle poles under the 100 x objective lens. We consider the spindle lateral with the coverslips if the two poles of spindle apparatus were on one focal plane. When the two poles were not on one focal plane, we defined it as 'non-lateral' with the coverslips.

To measure the spindle orientation or the spindle pole angles when they were exposed to 27 T magnetic fields that were parallel to the coverslips in *Figures 5–7*, we used the Leica DMI4000B fluorescent microscope to get low magnification images. Then we used the Picpick software to measure the angles between spindle long axis and the magnetic field lines on these images for *Figure 5* as well as the spindle pole angles in *Figure 7C*.

The spindle length, width, chromosome distribution, metaphase plate width and the angles between spindle long axis and the magnetic field lines in *Figures 6* and *7* were measured by Adobe Photoshop (RRID:SCR_014199). Spindles dimension and chromosome distributions were measured

from four independent coverslips of CNE-2Z cells and four independent coverslips of RPE1 cells that were treated with RO-3306 and CENP-E inhibitors from two independent assays.

## Cell synchronization

To increase the sample size and ensure strong statistics, we applied a standard drug synchronization protocol by using RO-3306 to arrest cells in G2 (*Vassilev, 2006*; *Vassilev et al., 2006*), then washout into MG132 to enrich cells in mitosis with intact spindles. We then collected many low magnification images to provide reliable statistics. RPE1 and CNE-2Z cells were plated onto the pre-cut coverslips 24 hr prior to the assay. Cells were then treated with RO-3306 (10 µM) for 21 hr to arrest cells at late G2 phase. After washing with warm PBS for three times, cells were subsequently treated with MG-132 (20 µM) or CENP-E inhibitor (GSK923295, 1 µM) for another 6 hr to arrest cells in metaphase or to produce misaligned chromosomes. Magnetic field was applied during the last 4 hr.

## Statistics

For quantifications in the manuscript, cells and spindles were counted for each condition from at least three to four independent coverslips or cell culture plates. Comparisons between treatments were analyzed by a two-tailed Student t test in GraphPad Prism software (RRID:SCR_002798). P values are labeled in the figures for where data were compared.

## Acknowledgement

We would like to thank Dr. Yuheng Zhang in the physical division of Chinese High Magnetic Field Laboratory for helpful discussions about the project. This work was supported by the National Key Research and Development Program of China (#2016YFA0400900), National Natural Science Foundation of China (Grant No U1532151), Hefei Science Center CAS (2016HSC-IU007) and Chinese Academy of Sciences 'Hundred Talent program' to Xin Zhang, National Natural Science Foundation of China (Grant Nos. 51627901, U1632160, 11374278) and the National Key Research and Development Program of China (#2016YFA0401003) to Qingyou Lu and Chinese High Magnetic Field Laboratory facility.

## Additional information

### Funding

| Funder | Grant reference number | Author |
| --- | --- | --- |
| National Key Research and Development Program of China | #2016YFA0400900 | Xin Zhang |
| National Natural Science Foundation of China | U1532151 | Xin Zhang |
| Hefei Science Center | 2016HSC-IU007 | Xin Zhang |

The funders had no role in study design, data collection and interpretation, or the decision to submit the work for publication.

### Author contributions

LZ, Conceptualization, Data curation, Validation, Investigation, Methodology, Writing—original draft, Writing—review and editing; YH, Investigation, Methodology, Writing—original draft, Writing—review and editing; ZL, Validation, Investigation, Methodology, Writing—review and editing; XJ, Validation, Investigation, Writing—original draft, Writing—review and editing; ZW, Investigation, Methodology, Writing—review and editing; HW, XT, Investigation, Writing—review and editing; FY, ZY, Data curation, Investigation, Writing—review and editing; LP, Methodology, Writing—review and editing; TJM, Supervision, Methodology, Writing—original draft, Writing—review and editing; QL, Supervision, Funding acquisition, Methodology, Writing—original draft, Writing—review and editing; XZ, Conceptualization, Supervision, Funding acquisition, Methodology, Writing—original draft, Writing—review and editing

Author ORCIDs

Xin Zhang, http://orcid.org/0000-0002-3499-2189

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
