## [Decision Letter]

Thank you for submitting your article "27 T Ultra-high Static Magnetic Field Changes Orientation and Morphology of Mitotic Spindles in Human Cells" for consideration by *eLife*. Your article has been reviewed by three peer reviewers, and the evaluation has been overseen by a Reviewing Editor and Anna Akhmanova as the Senior Editor. The following individuals involved in review of your submission have agreed to reveal their identity: Volker Hietschold (Reviewer #1); Nicolas Minc (Reviewer #2).

The reviewers have discussed the reviews with one another and the Reviewing Editor has drafted this decision to help you prepare a revised submission.

Summary:

As you can see, all three reviewers were intrigued by your work, and therefore we would like to publish your work in *eLife*, assuming you can address the two major concerns and some minor comments below.

Although the effects are convincing, it is not clear why smaller SMF has no effect, or in another word, why such a strong SMF must be applied to orient the spindle. What would help this part would be to relate the strength of the field to expected values of the torque exert based on estimation of DNA / MTs diamagnetism, needed to rotate the spindle. In addition, the dosage test of SMF strength is only performed for cells plated perpendicular to the SMF, and not using the assay in which the coverslip is parallel to the magnetic field, which is a much better measure of the studied effect. This should not be difficult to perform and should be added to the manuscript.

The experiments as well as the manuscript were often not as well organised as could be. The paper's structure is more like a description of development of ideas and history of the experiments. A lot of information about the questions to be answered, design of the experiments, cell types and so on is given in the results chapter – – – –. Parts of the discussion are also located in the results chapter – as well as additional questions –.

There also seemed to be some issues in independent variables. For instance, exposure times and magnetic field strengths were varied simultaneously. This should be corrected.

The comments of the reviewers are included below for your information. Please address the comments on data presentation, quantifications and statistical analyses.

Reviewer #2:

This manuscript by Zhang et al. addresses how cultured cells may orient their mitotic spindle to high steady magnetic fields (SMF). The authors make use of a sophisticated magnetic field incubation system, in which they place adherent human cells and assay the orientation of the spindle axis with the SMF.

The data are totally out of the beaten path, and provide in my opinion, very novel insights on the basic interactions between external signals and internal cellular structures. Beyond medical relevance, for instance to MRI, this study documents how molecular diamagnetic properties of different cellular components may cause large-scale effect at the cellular level. Overall, the paper is well written, and the significance of the data presented is trustworthy, although some effects appear to be small.

Reviewer #3:

In this paper, Zhang et al. use a 27 T static magnetic field to explore its effects to spindle orientation in vivo. Microtubules have been shown to align in vitro under magnetic fields, but this is the first time that spindle orientation under magnetic fields is studied. The authors mainly find that spindles orient perpendicular to the applied magnetic field, which is attributed to the chromosomes aligning parallel to the magnetic field, although the evidence for that is a bit unclear from Figure 6 (which is difficult to interpret and seems to suggest there is no change between wham and 27 T conditions). This evidence is consistent with the fact that perpendicularly aligned spindles have a slightly increased width as compared to parallel aligned spindles (obtained when chromosomes are misaligned). In summary, the aligning torque acting on chromosomes is larger than in spindles, resulting in the perpendicular alignment of spindles with respect to the magnetic field orientation. This also suggests that the chromosome-spindle interactions are strong enough to rotate the whole spindle. It would be interesting to know how these results change in larger spindles where the chromosome-spindle microtubule connections may be indirect. The ability of turning spindles is potentially interesting when used to investigate cleavage positioning during development.

---

## [Author Response]

*Summary:*

*As you can see, all three reviewers were intrigued by your work, and therefore we would like to publish your work in eLife, assuming you can address the two major concerns and some minor comments below.*

*Although the effects are convincing, it is not clear why smaller SMF has no effect, or in another word, why such a strong SMF must be applied to orient the spindle. What would help this part would be to relate the strength of the field to expected values of the torque exert based on estimation of DNA / MTs diamagnetism, needed to rotate the spindle. In addition, the dosage test of SMF strength is only performed for cells plated perpendicular to the SMF, and not using the assay in which the coverslip is parallel to the magnetic field, which is a much better measure of the studied effect. This should not be difficult to perform and should be added to the manuscript.*

This is a really good question. Since the magnetic torque depends on the magnetic field intensity and the magnetic susceptibility of the object so that it was not surprising to see multiple studies showing that the magnetic field strength is the key factor that determines the bioeffects of SMFs. For example, in 1993, Higashi et al. showed that 1 T SMF had only detectable alignment effect on erythrocytes while 4 T high SMF induced almost 100% alignment (Higashi et al. 1993). In 2011, Zhao et al. found that 8.5 T SMF could decrease the ATP level in AL cells but 1 T or 4 T SMFs could not (http://onlinelibrary.wiley.com/wol1/doi/10.1002/bem.20617/abstract. DOI: 10.1002/bem.20617). The torque of a substance is equal to the product of magnetic field intensity and the magnetic susceptibility of the object, in which the susceptibility could also be field-dependent and can be Taylor expanded. This means that the torque could be parabolically proportional to the magnetic field strength. Thus, a high field has much more severe impact than a lower field and this relationship is not linear. We have added this information to the discussion part of the revised manuscript.

For microtubules, it was proposed that the size of their diamagnetic anisotropy is such that 5 μm long microtubules could be completely aligned in a field of around 10 T (http://www.sciencedirect.com/science/article/pii/S0006349502754829; http://www.sciencedirect.com/science/article/pii/S0006349598778634). Valles et al. predicted that “it seems reasonable to presume that fields in excess of 10 T can align the MS”, which contains mainly microtubule and chromosomes. However, in their calculation, the contribution of chromosomes was not included, which is vertical to the spindle long axis so that they add an “antagonizing” effect. This alone indicates that it would require more than 10 T to rotate the spindle structure. Furthermore, based on the calculation in Maret G’s paper, 1.4 T SMF could align the mitotic chromosome (http://www.sciencedirect.com/science/article/pii/0921452690900778). When the chromosomes are combined with spindle microtubules, the final orientation of the whole spindle in the SMF will be determined by the balance between microtubules and the chromosomes. In addition, the whole spindle is also restricted inside the cell by aster microtubules as well as other surrounding factors, which make it even harder to rotate the spindle. Therefore it is not too surprising that the spindle orientation in cells need such a high SMF.

For the dosage test of SMF strength, we added a new set of experiment to test the effect of 1T and 9T on coverslips parallel to the magnetic field (New Figure 6—figure supplement 2)

*The experiments as well as the manuscript were often not as well organised as could be. The paper's structure is more like a description of development of ideas and history of the experiments. A lot of information about the questions to be answered, design of the experiments, cell types and so on is given in the results chapter. Parts of the discussion are also located in the results chapter as well as additional questions.*

Thanks for these suggestions about the writing. We have rearranged some parts of the manuscript to improve these points.

*There also seemed to be some issues in independent variables. For instance, exposure times and magnetic field strengths were varied simultaneously. This should be corrected.*

This is a good point. We added a new set of experiment to test the effect of 1 T and 9 T exposure for 4 hours, which is the same to the 27 T experiments, on coverslips parallel to the magnetic field (New Figure 6—figure supplement 2). We have also added a set of 9 T SMF exposure for 4 hours on coverslips vertical to the magnetic field (New Figure 4). For 9 T SMF, we did not do the 7-day treatment because 3-day is the maximum running time for that superconducting magnet. For the 27 T water-cooled magnet, the maximum running time is 4-5 hours. In contrast, for magnets below 1 T, since they are permanent magnets, there is no time limitation.

*Reviewer #3:*

*In this paper, Zhang et al. use a 27 T static magnetic field to explore its effects to spindle orientation* in vivo*. Microtubules have been shown to align* in vitro *under magnetic fields, but this is the first time that spindle orientation under magnetic fields is studied. The authors mainly find that spindles orient perpendicular to the applied magnetic field, which is attributed to the chromosomes aligning parallel to the magnetic field, although the evidence for that is a bit unclear from Figure 6 (which is difficult to interpret and seems to suggest there is no change between wham and 27 T conditions). This evidence is consistent with the fact that perpendicularly aligned spindles have a slightly increased width as compared to parallel aligned spindles (obtained when chromosomes are misaligned). In summary, the aligning torque acting on chromosomes is larger than in spindles, resulting in the perpendicular alignment of spindles with respect to the magnetic field orientation. This also suggests that the chromosome-spindle interactions are strong enough to rotate the whole spindle. It would be interesting to know how these results change in larger spindles where the chromosome-spindle microtubule connections may be indirect. The ability of turning spindles is potentially interesting when used to investigate cleavage positioning during development.*

Thanks for these comments. For Figure 6, the reviewer is right that the difference is not dramatic. Since bigger “a/b” value means better chromosome alignment, the up tilted red lines in the 27 T groups indicate that the spindles tend to orient perpendicular to the magnetic field direction (90^o^) when the chromosomes are well aligned at the metaphase plate.

Thanks a lot for these very informative suggestions. We are also interested in examining different types of spindles. We are currently collaborating with another group to study the development of *C. elegans* in high magnetic fields and got very promising results.